# Encounter complexes and dimensionality reduction in protein–protein association

**Dima Kozakov[1]\*, Keyong Li[2,3†], David R Hall[1†], Dmitri Beglov[1], Jiefu Zheng[2], Pirooz Vakili[3,4], Ora Schueler-Furman[5], Ioannis Ch Paschalidis[2,3]\*, G Marius Clore[6]\*, Sandor Vajda[1,7]\***

[1]Department of Biomedical Engineering, Boston University, Boston, United States; [2]Department of Electrical and Computer Engineering, Boston University, Boston, United States; [3]Division of Systems Engineering, Boston University, Boston, United States; [4]Department of Mechanical Engineering, Boston University, Boston, United States; [5]Department of Microbiology and Molecular Genetics, Hadassah Medical School, The Hebrew University of Jerusalem, Jerusalem, Israel; [6]Laboratory of Chemical Physics, National Institute of Diabetes and Digestive and Kidney Diseases, National Institutes of Health, Bethesda, United States; [7]Department of Chemistry, Boston University, Boston, United States

**Abstract** An outstanding challenge has been to understand the mechanism whereby proteins associate. We report here the results of exhaustively sampling the conformational space in protein–protein association using a physics-based energy function. The agreement between experimental intermolecular paramagnetic relaxation enhancement (PRE) data and the PRE profiles calculated from the docked structures shows that the method captures both specific and non-specific encounter complexes. To explore the energy landscape in the vicinity of the native structure, the nonlinear manifold describing the relative orientation of two solid bodies is projected onto a Euclidean space in which the shape of low energy regions is studied by principal component analysis. Results show that the energy surface is canyon-like, with a smooth funnel within a two dimensional subspace capturing over 75% of the total motion. Thus, proteins tend to associate along preferred pathways, similar to sliding of a protein along DNA in the process of protein-DNA recognition.

\*For correspondence: midas@ bu.edu (DK); vajda@bu.edu (SV); yannisp@bu.edu (ICP); mariusc@ intra.niddk.nih.gov (GMC)

†These authors contributed equally to this work

## Introduction

Interactions between proteins play a central role in various aspects of the structural and functional organization of the cell. To recognize its partner, a protein must align its binding interface, usually a small fraction of the total surface, with a similarly small binding interface on the other protein (*Berg and von Hippel, 1985*; *Ubbink, 2009*). Since all interactions are of relatively short range, the process must start with a diffusive search governed by Brownian motion, which brings the proteins to a 'macrocollision' to yield a transition state also known as the encounter complex (*Berg and von Hippel, 1985*; *von Hippel and Berg, 1989*). The encounter complex can be thought of as an ensemble of conformations in which the two molecules can rotationally diffuse along each other, or participate in a series of 'microcollisions' that properly align the reactive groups. The second step of association consists of conformational rearrangements to the native complex. While it has been generally recognized that association proceeds through a transition state, little was known of the encounter complex structures and configurations as their populations are low, their lifetimes are short, and they are difficult to trap, rendering them essentially invisible to conventional structural and biophysical methods (*Iwahara and Clore, 2006*).

Novel experimental and improved computational methods, developed during the last decade, have the potential to provide information leading to better understanding of the nature of encounter

**eLife digest** Proteins rarely act alone. Instead, they tend to bind to other proteins to form structures known as complexes. When two proteins come together to form a complex, they twist and turn through a series of intermediate states before they form the actual complex. These intermediate states are difficult to study because they don't last for very long, which means that our knowledge of how complexes are formed remains incomplete.

One promising approach for studying the formation of complexes is called paramagnetic relaxation enhancement. In this technique certain areas in one of the proteins are labelled with magnetic particles, which produce signals when the two proteins are close to each other. Repeating the measurement several times with the magnetic particles in different positions provides information about the overall structure of the complex. Computational modelling can then be used to work out the fine details of the structure, including the shapes of the intermediate structures made by the proteins as they interact.

A computer method called docking can be used to predict the most favourable positions that the proteins can take, relative to one another, in a complex. This involves calculating the energy contained in the system, with the correct structure having the lowest energy. Docking methods also predict protein models with slightly higher energies, but with structures that are radically different. Modellers usually ignore these structures, but comparing the docking results to paramagnetic relaxation enhancement data, Kozakov et al. found that these structures actually represent the intermediate states.

Analysing the structure of the intermediate states revealed that the movement of the two proteins relative to one another is severely restricted as they form the final complex. Kozakov et al. found that proteins associate along preferred pathways, similar to the way a protein slides along DNA in the process of protein-DNA recognition. Knowing that the movement of the proteins is restricted in this way will enable researchers to improve the efficiency of docking calculations.

complexes. On the experimental side, the major progress is due to the application of NMR paramagnetic relaxation enhancement (PRE), a technique that is exquisitely sensitive to the presence of lowly populated states in the fast exchange regime (*Clore, 2008*; *Clore and Iwahara, 2009*; *Fawzi et al., 2010*). The detection of such intermediates requires introducing paramagnetic labels, one at a time, at a few sites on one of the interacting proteins, and measuring the transverse paramagnetic relaxation enhancement (PRE) rates, $\Gamma_2$, of the backbone amide protons ($^1H_N$) of the partner protein (*Tang et al., 2006*). In a fast exchanging system, the observed value of $\Gamma_2$ is the weighted average of the values for the various states present in solution (*Iwahara et al., 2004*; *Iwahara and Clore, 2006*). Because $\Gamma_2$ is dependent on the inverse sixth power of the distance ($<r^{-6}>$) between the unpaired electron on the paramagnetic center and the observed proton, and because the $\Gamma_2$ rates at short distances are very large owing to the large magnetic moment of the unpaired electrons, low-population intermediates can be detected. In particular, the observed intermolecular $^1H_N-\Gamma_2$ rates and those back-calculated from the structure of the native complex generally differ for a number of residues, revealing regions that participate in transitional interactions (*Tang et al., 2006*).

Paramagnetic relaxation enhancement (PRE) techniques can provide distributions of distances between a paramagnetic ion and protons, indicating the presence and relative ratio of conformational sub-ensembles (*Tang et al., 2006*; *Suh et al., 2007*; *Clore and Iwahara, 2009*; *Fawzi et al., 2010*), but determining the detailed structure of encounter complexes requires computational approaches. A semi-quantitative depiction of the minor species can be obtained by using restrained rigid-body simulated annealing refinement to minimize the difference between observed and calculated $^1H_N-\Gamma_2$ rates (*Tang et al., 2006*; *Kim et al., 2008*). However, with the development of docking methods it is possible to globally sample the entire conformational space of two interacting proteins, generating all low energy states. Although in some cases binding is inherently coupled with folding (*Shoemaker et al., 2000*; *Zheng et al., 2012*), a large class of protein complexes can be adequately described by a model that assumes essentially rigid association, possibly followed by refinement that allows for local changes in side chains and interacting loops (*Smith and Sternberg, 2002*; *Kozakov et al., 2006*; *Vajda and Kozakov, 2009*). As demonstrated by the results of CAPRI (Critical Assessment of Prediction

of Interactions) communitywide protein docking experiment, for such cases modern computational docking methods, including automated servers, are capable of generating docked conformations that agree well with the X-ray structure of the complex (*Lensink and Wodak, 2013*). In particular, our program PIPER, based on the Fast Fourier transform (FFT) correlation approach, globally and systematically samples the conformational space of two interacting proteins on a dense grid using a physics based energy function (*Kozakov et al., 2006*). The program is implemented in the heavily used server ClusPro (*Comeau et al., 2007*), which yields good results when docking X-ray structures of two proteins with at most moderate backbone conformational changes upon binding (*Kozakov et al., 2010*). Based on the results of CAPRI, Cluspro has been the best protein–protein docking server for the last 5 years (*Kozakov et al., 2013*; *Lensink and Wodak, 2013*).

It is well known that, in addition to near-native structures, docking generally yields a large number of models that are similar to near-native ones in terms of energy, but may substantially differ in terms of geometry (*Vajda and Kozakov, 2009*). Since such 'false positive' models do not predict the final bound complex, they are usually regarded as artifacts. However, using molecular mechanics energy functions without 'built-in' information on the native state it is reasonable to assume that the alternative low energy models represent encounter complexes that, in view of their favorable interactions, may occur along association pathways. Accordingly, we show here that using the large ensemble of low energy structures generated by docking provides better approximation of experimental PRE profiles than the one calculated only from the coordinates of the final complex. Since in docking calculations we start from unbound protein structures and systematically sample the entire conformational space, based on this result we can easily generate ensembles of encounter complexes for any pair of associating proteins.

Once it is established that the energy function used for sampling the conformational space enables us to accurately predict both the native state and the ensemble of encounter complexes, and thus the energy function is valid beyond selecting the native structure of the complex, we proceed to characterizing the energy landscape in the 6D translational/rotational space near the native state (*McCammon, 1998*; *Camacho et al., 1999*; *Zhang et al., 1999*; *Tovchigrechko and Vakser, 2001*). We focus on the main binding funnel in a neighborhood of the native state, which is the most important region of the conformational space, containing over 90% of complex structures observed in the PRE experiments (*Iwahara and Clore, 2006*; *Tang et al., 2006*; *Clore and Iwahara, 2009*; *Fawzi et al., 2010*). Since rigid body association occurs in a low dimensional space, the shape of the binding energy landscape can be studied in detail, in contrast to protein folding, which occurs in a very high dimensional space (*Dill and Chan, 1997*). In spite of the low dimensionality, the analysis is far from simple in this highly curved space due to the interdependence of the coordinates (*Park, 1995*; *Park and Ravani, 1997*; *Shen et al., 2008*; *Mirzaei et al., 2012*). However, one can transform the rotational space into a product of axis-angle representations using complex exponentials (*Park and Ravani, 1997*). These so-called exponential parameters create a local one-to-one mapping between the nonlinear manifold of potential conformations and an Euclidean space (*Mirzaei et al., 2012*), and thus the shape of low energy regions in the conformational space can be studied by classical principal component analysis (PCA) in the Euclidean space. Using PCA we will be able to determine whether any subspace can accommodate a large fraction of the structures, and whether there are energy barriers that restrict the distribution of encounter complexes in the vicinity of the native state.

The most important result of our analysis is that the region of the space in a neighborhood of the native state invariably includes high energy barriers preventing the ligand from moving into a one- or two-dimensional restrictive subspace. Orthogonal to the restrictive subspace is a permissive subspace, in which the energy is relatively flat. Based on these results one can visualize the energy landscape as resembling a canyon-like terrain where the low energy areas (at the bottom of the canyon) lie in a lower dimensional subspace. Thus, within the range of physical interactions, two proteins sample only relatively small fractions of the conformational space, and converge toward the native state along preferred pathways. This result represents information that, in principle, could have been obtained by running molecular dynamics or Brownian dynamics simulations. However, a sufficiently dense sampling of conformational space by molecular dynamics is computationally very demanding, even when restricting considerations to a neighborhood of the native state (*Wang and Wade, 2003*), whereas Brownian dynamics simulations usually rely on highly simplified protein models (*Camacho et al., 2000*; *Spaar and Helms, 2005*). In contrast, we use detailed all-atom models and map the energy surface using two different physics-based energy functions. As will be shown, these differences do

not significantly affect the results of the principal component analysis, suggesting that the reduction of dimensionality is an inherent property of the free energy landscape in protein–protein association.

## Results

### Prediction of encounter complex ensembles

We focused on the modeling of the association between the N-terminal domain of Enzyme I (EIN) and the histidine-containing phosphocarrier protein (HPr) (*Figure 1A*), because the complex has been studied in a series of PRE titration experiments (*Fawzi et al., 2010*). Specific association between the two proteins occurs in the first step of the bacterial phosphotransfer system, resulting in phosphoryl transfer between EIN and HPr upon proper alignment of active site histidines of the two sides of the interface (*Garrett et al., 1999*). The binding has an equilibrium dissociation constant of 4.3 µM (*Suh et al., 2008*). For the computational study of encounter complexes we have placed the center of EIN at the origin of a coordinate system, and systematically sampled the entire rotational/translational space of HPr. Unbound structures were used both for the receptor, EIN (chain A from PDB entry 1ZYM)

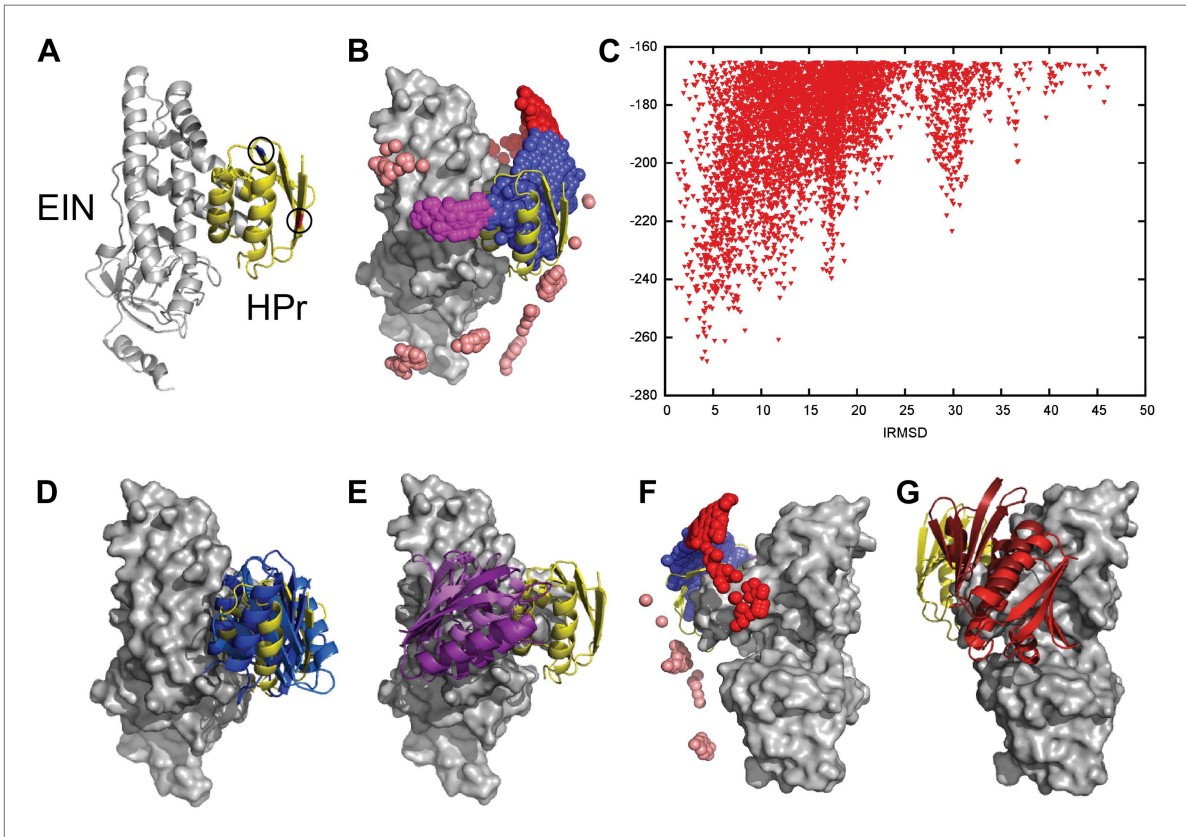

**Figure 1**. Docking results for the EIN–HPr complex. Unbound structures were used both for the receptor, EIN (chain A from PDB entry 1ZYM) and for the ligand, HPr (chain P from PDB entry 2JEL). Encounter complexes were generated using Fast Fourier transform (FFT) based sampling. (**A**) Cartoon of the specific complex formed by EIN and HPr, shown in grey and yellow, respectively. The locations of the paramagnetic tags E5C-EDTA-Mn+ and E32C-EDTA-Mn$^{2+}$ on HPr are encircled and are shown in red and blue, respectively. (**B**) Centers of HPr structures in the encounter complex ensemble. Colors indicate classification as follows (*8*): blue, Class I (i.e., overlapping with the specific complex); magenta, patch 1 of Class II (i.e., non-overlapping) positions; red, patch 2 of Class II positions; and pink, additional Class II position outside the main patches. (**C**) Ligand IRMSD vs PIPER energy score. (**D**) Two representative HPr poses, colored light blue and dark blue, from Class I. (**E**) Two representative HPr poses (in different shades of magenta) from Patch 1 of Class II. (**F**) View of the EIN–HPr complex and the centers of HPr poses after rotating 180° around the vertical axis (the bound HPr is now on the left side, almost completely hidden by EIN). (**G**) Representative HPr poses (in different shades of red) from Patch 2 of Class II, shown in the rotated view.

The following figure supplements are available for figure 1:

**Figure supplement 1**. Rotamers of the paramagnetic labels E5C-EDTA-Mn$^{2+}$ and E32C-EDTA-Mn$^{2+}$ on HPr.

and for the ligand, HPr (chain P from PDB entry 2JEL). Sampling was performed using the docking program PIPER, which performs exhaustive evaluation of a physics-based energy function in discretized 6D space of mutual orientations of two proteins using the Fast Fourier transform (FFT) correlation approach (*Kozakov et al., 2006*). We sample 70,000 rotations, which approximately correspond to sampling at every 5° in the space of Euler angles. In the translational space, the sampling is defined by the 1.0 Å grid cell size. PIPER is used with a 'smooth' energy function that includes terms describing attractive and repulsive van der Waals interactions, electrostatic interactions calculated by a simplified generalized Born-type expression, and a desolvation terms, the latter represented by a pairwise interaction potential (*Chuang et al., 2008*). We call the energy function 'smooth' because the repulsive contributions to the van der Waals interaction are selected to allow for a certain amount of overlaps.

Since the generated structures will be used for calculating PRE profiles to compare them to experimental PRE data (*Fawzi et al., 2010*), two sets of docking calculations were performed using HPr structures that included the paramagnetic label EDTA-$Mn^{2+}$, placed either at E5C, which is distal to the EIN/HPr interface in the native state, or at E32C, which is close to the edge of the interface (*Figure 1A*). Each C-EDTA-$Mn^{2+}$ label has three potential rotameric states, and hence $Mn^{2+}$ can occupy three different positions (*Figure 1—figure supplement 1*). A separate docking was performed for each rotameric state of each EDTA-$Mn^{2+}$ probe. We retained the 10,000 lowest energy structures from each docking simulation, thus a total of 30,000 low energy structures for each of the two probes. These structures were then used for the analysis of encounter complexes and for the calculation of intermolecular PRE rates.

*Figure 1B* shows the center of each low energy HPr structure, generated by the docking, as a small sphere, and indicates that these structures form three major clusters. *Figure 1C* shows the Interface Root Mean Square Deviation (IRMSD) from the native complex vs the PIPER energy score of the docked structures. For the calculation of IRMD we first select the interface residues of HPr that are within 10 Å of any EIN atom in the native complex. For each docked structure we than superimpose EIN onto EIN in the X-ray structure of the complex, and calculate the RMSD between the Cα atoms of the HPr interface residues in docked and native structures. The structures in the largest cluster (shown in blue in *Figure 1B*) overlap with the native state, with the lowest energy conformations being within 5 Å IRMSD from the native (*Figure 1C*). The structures in this cluster, termed Class I, are the results of rigid body rotations and small translations around the native binding mode. Two representative Class I structures are shown in *Figure 1D*. Some Class I structures have less than 1 Å IRMSD, but the cluster extends as far as 15 Å IRMSD from the native. The two other clusters, termed Class II patch 1 and Class II patch 2, consist of structures that can coexist with the native complex. We note that while the three clusters clearly separate in the 3D representation shown in *Figure 1B*, they substantially overlap when projected into one dimension as a function of their IRMSD values. Nevertheless, *Figure 1C* shows at least three distinguishable energy minima. Class II patch 1 (magenta in *Figure 1B*) centers around a local energy minimum at around 17 Å IRMSD (*Figure 1C*). *Figure 1E* shows two representative conformations for this patch. The third large cluster, Class II patch 2 (red in *Figure 1B*), is located on the opposite side of the Class I cluster, and is better seen after rotating the complex by 180° around its vertical axis (*Figure 1F*). The local energy minimum in this cluster is located at about 30 Å IRMSD from the native state. *Figure 1G* shows two representative conformations for the Class II patch 2. In addition to the complexes that belong to Class I and the two patches of Class II, there are a number of smaller patches, shown in pink in *Figure 1B*.

## PRE experiments and theoretical PRE profiles based on structure

To detect encounter complexes in the EIN/HPr system by PRE titration experiments, HPr was labeled with a paramagnetic EDTA-$Mn^{2+}$ moiety conjugated via a disulfide bond to surface cysteine mutations at specific sites (*Fawzi et al., 2010*). We consider the mutations E5C and E32C that are both located outside the specific interaction surface with EIN (*Figure 1A*) and thus the labels do not interfere with the formation of the native complex. Intermolecular $^{1}H_N$–$\Gamma_2$ rates for the backbone amide protons of U-[$^{2}$H,$^{15}$N]-labeled EIN were measured in the presence of 150 mM NaCl to eliminate potential spurious nonspecific interactions not relevant at physiological ionic strength. PRE measurements were carried out at six different concentrations of the paramagnetically labeled HPr (ranging from 60 to 450 μM), corresponding to HPr:EIN molar ratios of 0.2–1.5. At each point in the titration, the intermolecular PREs were summed over their respective residues and normalized to the highest value of each titration curve. The data points in *Figure 2* show the normalized intermolecular PRE values and their standard errors observed in these titration experiments (*Fawzi et al., 2010*).

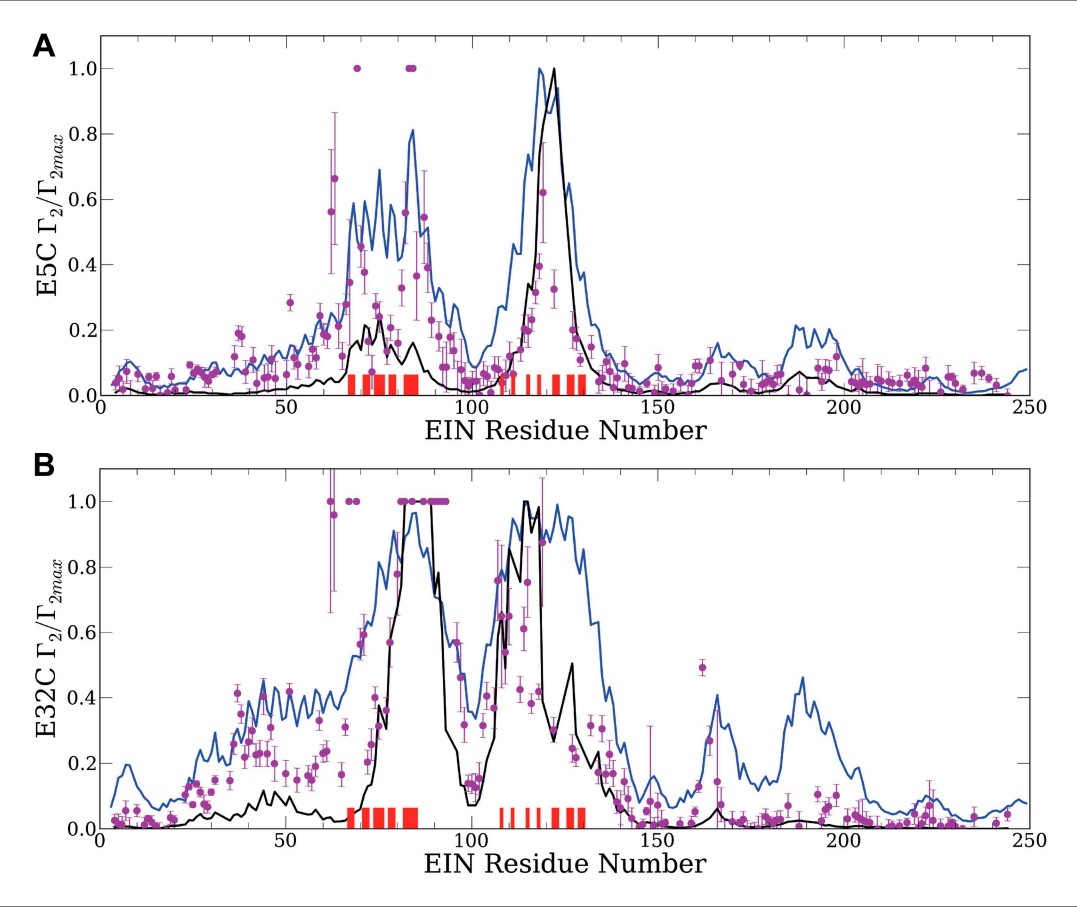

**Figure 2**. Normalized intermolecular PRE profiles for the EIN–HPr complex. PRE measurements were carried out at 300 µM EIN, 300 µM HPr, and 150 mM NaCl (*Fawzi et al., 2010*). Theoretical intermolecular PREs, calculated only from the coordinates of the specific EIN/HPr complex, are shown as black lines. Calculated PRE values, based on all generated encounter complexes, are shown as blue lines, and reveal substantial contributions by the non-specific structures. The experimental PRE rates ($\Gamma_2$) are displayed as filled-in magenta circles. Points representing $\Gamma_2$ values that were too large (>60 s$^{-1}$) to be determined accurately are placed at the saturation level $\Gamma_2/\Gamma_{2max} = 1$. Interface residues are indicated by red ticks on the x-axis. (**A**) Results for EIN/HPr-E5C-EDTA-Mn$^{2+}$ complexes. (**B**) Results for EIN/HPr-E32C-EDTA-Mn$^{2+}$ complexes.

The following figure supplements are available for figure 2:

**Figure supplement 1**. Controls emphasizing the need for accurate energy function in docking: theoretical PRE profiles for the EIN/HPr complex, based on complexes generated by using only the van der Waals energy (blue line).

**Figure supplement 2**. Normalized intermolecular PRE profiles and encounter compexes for the IIA$^{Mannitol}$/HPr interactions.

**Figure supplement 3**. Normalized intermolecular PRE profiles and encounter compexes for the HPr/HPr interactions.

**Figure supplement 4**. Encounter complexes in the Cytochrome c–Cytochrome c peroxidase interactions as reported on the basis of PRE experiments (*Bashir et al., 2010*), shown as pink spheres, and the ones generated by the PIPER docking program, shown as blue spheres.

---

Given the coordinates of a complex, one can back-calculate theoretical PRE profiles ('Materials and methods'). As shown in *Figure 2*, the theoretical profile calculated from the coordinates in the X-ray structure of the native complex (black curve) substantially deviates from the experimental values for a number of residues. For HPr-E5C the largest differences occur at positions 59–97 and 160–167 of EIN,

with smaller deviations at 23–37 and 183–189 (*Figure 2A*). For HPr-E32C the large differences are at positions 22–51, 56–74, 96–106, and 160–167 of EIN, with smaller differences at 184–189 (*Figure 2B*). These deviations show that the observed PRE rates cannot be explained well on the basis of the native binding mode of HPr alone, and provide at least qualitative evidence for the existence of lowly populated encounter states in rapid exchange with the final native complex (*Tang et al., 2008*).

Our hypothesis is that the non-native structures generated by the docking represent encounter complexes, and hence accounting for these structures predicts the experimental PRE values better than the profile calculated only from the coordinates of the native complex. Therefore we calculate the theoretical PRE profiles based on all 30,000 low energy structures obtained by the docking for each position of the paramagnetic label (blue lines in *Figure 2*). The details of the calculation are given in the 'Materials and methods'. We emphasize that these curves are based only on the docked structures, and the native binding mode in the X-ray structure of the complex is not used. The results show that the hypothesis is certainly true for EIN/HPr-E5C, because the full encounter ensemble provides much better approximation than the native complex, particularly for residues 59–97 and 160–167 (*Figure 2A*). The correlation coefficient between the experimental PRE rates and the ones based on the encounter complexes is 0.705. In contrast, the correlation coefficients between the experimental PRE rates and the ones back-calculated from the native structure (black line in *Figure 2A*) is only 0.47. We note that the agreement improves even for the interface residues in the 67–85 region of EIN (indicated by red ticks in *Figure 2*). The explanation is that the E5C-EDTA-$Mn^{2+}$ label is on the far side of HPr from the interface, and in some encounter complexes this label is much closer to interface residues than in the native complex. For HPr with the E32C-EDTA-$Mn^{2+}$ label, accounting for the encounter complexes generated by docking (blue curve in *Figure 2B*) improves the correlation coefficient more moderately, from 0.709 to 0.77. This is due to the fact that E32C is close to the edge of the interface in the native complex, already providing a strong PRE signal for residues 67–85, resulting in a high correlation coefficient with the experimental PRE values. As will be discussed, since the PRE data are sensitive to small conformational changes and thus are inherently noisy, it is difficult to further improve an already high correlation coefficient. However, even for HPr-E32C, the PRE profile back-calculated from the low energy models still yields better prediction than considering only the native binding mode. Since it is well known that the PRE profiles heavily depend on the location of the paramagnetic tag relative to the native interface (*Fawzi et al., 2010*), this result does not contradict to our hypothesis that accounting for all structures generated by the docking improves the prediction of PRE rates.

To provide a control and to demonstrate that the use of an accurate energy function is very important for generating a meaningful ensemble of encounter complexes we have also performed docking calculations using a scoring function without long-range energy terms, that is, considering only the attractive and repulsive components of the van der Waals energy. This simplified energy function yields docked structures that have good shape complementarity, but have no favorable electrostatic or chemical interactions. The 30,000 structures with the lowest van der Waals energy from this 'shape-complementarity only' docking were then used for back-calculating theoretical PRE profiles. The results of these calculations, shown in *Figure 2—figure supplement 1*, make it absolutely clear that the back-calculated PRE profiles based on the ensemble of structures generated without a proper energy function do not show any resemblance to the observed PRE data. In fact, both correlation coefficients between theoretical and experimental PRE rates are negative, −0.36 and −0.58, respectively, for the probes at positions E5C and E32C.

Considering encounter complexes generated by PIPER using its physics based energy function we also obtained good agreement with experimental PRE data for other pairs of proteins. The first is the IIA$^{Mannitol}$/HPr complex (*Cornilescu et al., 2002*; *Tang et al., 2006*). *Figure 2—figure supplement 2,B* show the native structure of the complex and the ensemble of docked structures generated by PIPER. The paramagnetic label is placed at E5C of HPr, colored red and indicated by a small circle in *Figure 2—figure supplement 2A*. It is important that, similarly to the EIN/HPr system, the E5C-EDTA-$Mn^{2+}$ label is on the far side of HPr from the interface in the native structure of the complex. *Figure 2—figure supplement 2C* shows the experimental PRE data, the theoretical PRE profile based on the native complex (black line), and the theoretical profile obtained by considering the 30,000 low energy structures generated by the docking (blue line). The correlation coefficient between the experimental PRE rates and the ones back-calculated from the native structure is 0.58, whereas using the docked structures for the PRE calculation increases the correlation coefficient to 0.78, demonstrating substantially improved prediction. Although some of the improvements occur at the interface residues, it is clearly

helpful that the HPr-E5C-EDTA-Mn$^{2+}$ label is far from the interface. The distance between the label and a number of IIA$^{Mannitol}$ residues is substantially reduced in some of the encounter complexes, which makes the presence of minor species more pronounced. We also show observed PRE data and theoretical profiles calculated from the ensemble of structures generated by docking for the complexes HPr/HPr (*Tang et al., 2008*; *Figure 2—figure supplement 3*), and cytochrome c/cytochrome c peroxidase (*Bashir et al., 2010*) (*Figure 2—figure supplement 4*), demonstrating good qualitative agreement in both cases.

## Energy landscapes of encounter complexes

Having established that the sampling algorithm and energy function are accurate enough for predicting ensembles of encounter complexes, we proceeded to the characterization of the energy landscape in a neighborhood of the native complex conformation. As in the previous section, we focused on the rigid-body motions of the ligand protein in the space fixed on the receptor protein, although local structural adjustments of the proteins were allowed for more accurate energy calculation. Geometrically the 6D translational/rotational space is the so-called Special Euclidean Group *SE(3)*, which is the semidirect product of $R^3$ of the translations and *SO(3)* of the rotations (*Park, 1995*; *Park and Ravani, 1997*; *Shen et al., 2008*; *Mirzaei et al., 2012*). Restricting considerations to encounter complexes in which the surfaces of the two proteins touch each other removes the distance of the two proteins as a variable, and the space can be parameterized in terms of 5 angular coordinates. Two angles are needed to define the direction from the center of the receptor to the center of the ligand interface, and the other three angles specify the rotation of the ligand. Although the resulting space is nonlinear and thus the 5 angular coordinates are interdependent, a 5D Euclidean space can be mapped onto this nonlinear space using exponential maps (*Shen et al., 2008*), and hence we will be able to use analysis tools such as PCA, developed for application in Euclidean space. Further details justifying the need for the use of exponential maps will be given in 'Materials and methods'. Once an appropriate coordinate system was defined, we selected and densely sampled a region in the neighborhood of the native state to obtain information on the shape of the binding funnel (*Camacho et al., 1999*; *Selzer and Schreiber, 2001*; *Wang and Wade, 2003*; *Miyashita et al., 2004*). Since the apparent properties of the landscape depend both on the energy evaluation model and the method of sampling, to assess the generality of the results we used both PIPER (*Kozakov et al., 2006*) and the very different docking program RosettaDock (*Gray et al., 2003*), which is based on Monte Carlo minimization and rebuilds side chain conformations during the search. From each sampling calculation, performed either by PIPER or by RosettaDock, we selected the conformations below a certain energy threshold to delineate the floor of the energy funnel.

Encounter complexes were generated from unbound protein structures (*Chen et al., 2003*) for a diverse set of 42 interacting protein pairs (*Table 1*). For each of these complexes, selected from the protein docking benchmark (*Chen et al., 2003*), both PIPER and RosettaDock found an energy funnel near the native state. Since this is generally not the case for complexes involving multiple subunits or large conformational changes upon binding, such complexes in the benchmark set were not considered. Structures were retained within 10 Å IRMSD from the native state. After sampling, the exponential coordinates were normalized to ensure that the variances in the sample set are the same along each coordinate axis, and the shape of the energy landscape over the selected region was studied by applying principal component analysis (PCA) to 5% of conformations with the lowest energy values. As will be emphasized in 'Materials and methods', the use of exponential maps, resulting in independent coordinates, is crucial for the success of our study, as only in this case can PCA separate the essential hyperspaces that bound the low energy ensemble.

The eigenvalues obtained by PCA are normalized to add to 100%. Each eigenvalue $\lambda_i$ can be then interpreted as the percentage of the total variance that is accounted for by the variance along the corresponding eigenvector $v_i$. The smallest eigenvalue, $\lambda_5$, is less than 5% for almost all complexes (*Table 1*). In many cases both $\lambda_5$ and $\lambda_4$ are small (their sum is less than 10%), indicating that the eigenvectors $v_4$ and $v_5$ span a 'restrictive' subspace where the low energy structures barely deviate from the native complex. In contrast, $\lambda_1$ and $\lambda_2$ typically sum up to more than 75% of variance. Thus, it is expected that in the 'permissive' subspace spanned by $v_1$ and $v_2$ the low energy structures may substantially differ from the native conformation.

As an example, *Figure 3A* shows IRMSD and energy distributions along the five eigenvectors, calculated from the low energy structures generated by PIPER, for the complex between the retinoid

**Table 1.** Eigenvalues (in %) obtained by PCA, and the angle between restrictive subspaces

| PDB ID | PIPER | | | | | RosettaDock | | | | | Discrepancy (degrees) |
|---|---|---|---|---|---|---|---|---|---|---|---|
| | $\lambda_1$ | $\lambda_2$ | $\lambda_3$ | $\lambda_4$ | $\lambda_5$ | $\lambda_1$ | $\lambda_2$ | $\lambda_3$ | $\lambda_4$ | $\lambda_5$ | |
| 1AVX | 59.4 | 32.8 | 6.2 | 1.2 | 0.3 | 67.4 | 15.5 | 13.4 | 3.4 | 0.3 | 5 |
| 1B6C | 72.1 | 19.1 | 6.9 | 1.3 | 0.5 | 84.2 | 10.2 | 3.3 | 1.9 | 0.4 | 4 |
| 1 E6E | 59.1 | 18.1 | 11.3 | 10.0 | 1.5 | 57.6 | 16.3 | 10.9 | 8.8 | 6.4 | 29 |
| 1EAW | 44.3 | 31.7 | 22.2 | 1.0 | 0.9 | 57.9 | 33.0 | 4.6 | 4.0 | 0.4 | 25 |
| 1 E6J | 78.7 | 13.5 | 7.1 | 0.3 | 0.3 | 47.6 | 31.5 | 18.7 | 1.2 | 1.0 | 16 |
| 1GLA | 58.9 | 26.9 | 9.3 | 3.7 | 1.2 | 41.5 | 33.1 | 15.1 | 8.0 | 2.3 | 2 |
| 1IQD | 74.7 | 13.5 | 7.8 | 3.6 | 0.4 | 58.0 | 26.7 | 12.7 | 1.9 | 0.7 | 13 |
| 1K74 | 47.8 | 28.0 | 19.0 | 3.6 | 1.5 | 61.0 | 22.0 | 10.5 | 5.2 | 1.2 | 19 |
| 1MAH | 60.3 | 22.4 | 11.7 | 4.4 | 1.2 | 52.8 | 22.0 | 13.5 | 7.7 | 4.0 | 14 |
| 1N8O | 56.1 | 23.4 | 13.3 | 6.4 | 0.9 | 66.9 | 22.3 | 10.3 | 0.3 | 0.2 | 20 |
| 1PPE | 56.4 | 26.4 | 14.9 | 1.7 | 0.6 | 47.1 | 44.4 | 7.9 | 0.4 | 0.1 | 3 |
| 1PXV | 68.3 | 17.0 | 9.6 | 4.3 | 0.8 | 32.1 | 27.2 | 23.6 | 14.5 | 2.7 | 8 |
| 1 R0R | 55.0 | 26.8 | 15.3 | 2.7 | 0.2 | 69.3 | 20.2 | 6.7 | 3.0 | 0.9 | 13 |
| 2SNI | 49.3 | 31.6 | 17.0 | 1.5 | 0.6 | 79.6 | 15.5 | 3.5 | 1.2 | 0.2 | 16 |
| 1KXQ | 47.7 | 30.0 | 16.8 | 4.1 | 1.3 | 66.3 | 30.0 | 4.4 | 0.2 | 0.1 | 29 |
| 7CEI | 44.7 | 28.5 | 20.9 | 4.6 | 1.3 | 47.7 | 27.9 | 18.9 | 3.6 | 1.9 | 19 |
| 2SIC | 58.6 | 23.4 | 9.4 | 7.2 | 1.4 | 84.2 | 8.8 | 4.1 | 2.4 | 0.5 | 3 |
| 1AY7 | 56.9 | 20.3 | 15.0 | 5.4 | 2.4 | 42.1 | 32.5 | 12.6 | 9.2 | 3.7 | 27 |
| 1OPH | 72.6 | 15.5 | 9.2 | 2.2 | 0.5 | 84.2 | 9.3 | 5.9 | 0.4 | 0.2 | 21 |
| 1UDI | 64.6 | 18.6 | 12.6 | 2.4 | 1.8 | 51.2 | 27.1 | 14.2 | 6.4 | 1.1 | 33 |
| 1BUH | 44.8 | 27.7 | 17.6 | 9.2 | 0.6 | 40.3 | 32.9 | 16.6 | 8.2 | 1.9 | 21 |
| 1FSK | 45.0 | 28.0 | 22.1 | 3.5 | 1.4 | 42.6 | 29.7 | 19.9 | 5.8 | 1.9 | 21 |
| 1JPS | 57.1 | 25.7 | 12.4 | 4.0 | 0.8 | 56.3 | 28.8 | 13.7 | 0.7 | 0.6 | 30 |
| 1DQJ | 51.4 | 31.3 | 15.0 | 1.4 | 0.9 | 46.5 | 19.8 | 17.4 | 12.3 | 4.0 | 17 |
| 2B42 | 55.6 | 27.7 | 12.8 | 3.4 | 0.5 | 45.4 | 23.1 | 15.7 | 11.5 | 4.4 | 24 |
| 2FD6 | 65.1 | 18.1 | 9.9 | 4.6 | 2.2 | 36.4 | 23.9 | 21.0 | 13.7 | 5.0 | 20 |
| 2HQS | 80.1 | 11.3 | 7.2 | 1.0 | 0.4 | 54.1 | 36.2 | 7.8 | 1.6 | 0.4 | 9 |
| 2I25 | 70.3 | 18.5 | 9.9 | 0.8 | 0.5 | 56.0 | 15.3 | 12.8 | 10.5 | 5.3 | 12 |
| 2MTA | 45.8 | 26.1 | 20.5 | 5.2 | 2.5 | 45.5 | 32.2 | 12.2 | 7.4 | 2.7 | 30 |
| 1MLC | 59.3 | 31.5 | 6.8 | 1.2 | 1.1 | 42.7 | 30.1 | 17.3 | 7.0 | 2.8 | 17 |
| 2HRK | 57.7 | 31.0 | 9.7 | 0.9 | 0.7 | 61.2 | 16.4 | 10.5 | 8.6 | 3.3 | 30 |
| 1AHW | 74.4 | 16.3 | 7.7 | 1.1 | 0.5 | 46.7 | 31.5 | 17.2 | 3.1 | 1.5 | 24 |
| 1Z5Y | 66.1 | 18.0 | 8.9 | 5.4 | 1.5 | 57.1 | 32.3 | 9.7 | 0.6 | 0.4 | 29 |
| 2HLE | 54.0 | 28.1 | 13.0 | 3.4 | 1.4 | 67.0 | 12.9 | 12.1 | 5.9 | 2.1 | 2 |
| 2NZ8 | 69.4 | 14.1 | 8.4 | 5.2 | 2.9 | 44.8 | 28.3 | 15.1 | 7.3 | 4.5 | 34 |
| 1BVN | 61.4 | 20.3 | 14.3 | 3.6 | 0.4 | 37.2 | 28.5 | 17.8 | 10.0 | 6.5 | 32 |
| 1CGI | 66.9 | 15.4 | 10.9 | 5.4 | 1.9 | 57.9 | 27.9 | 10.7 | 3.0 | 0.5 | 48 |
| 1GPW | 53.8 | 20.9 | 14.2 | 6.4 | 4.7 | 46.5 | 26.5 | 17.9 | 5.0 | 4.0 | 27 |
| 2JEL | 72.4 | 16.3 | 8.1 | 2.2 | 0.9 | 47.8 | 31.6 | 12.9 | 6.8 | 0.8 | 31 |
| 1NCA | 76.7 | 18.8 | 3.0 | 1.2 | 0.4 | 55.9 | 24.3 | 15.6 | 3.2 | 1.0 | 27 |
| 2UUY | 69.4 | 15.4 | 12.7 | 1.6 | 0.9 | 73.6 | 16.2 | 9.0 | 1.0 | 0.2 | 11 |
| 1KAC | 48.9 | 34.6 | 13.3 | 2.0 | 1.2 | 53.8 | 19.5 | 15.6 | 6.8 | 4.2 | 30 |

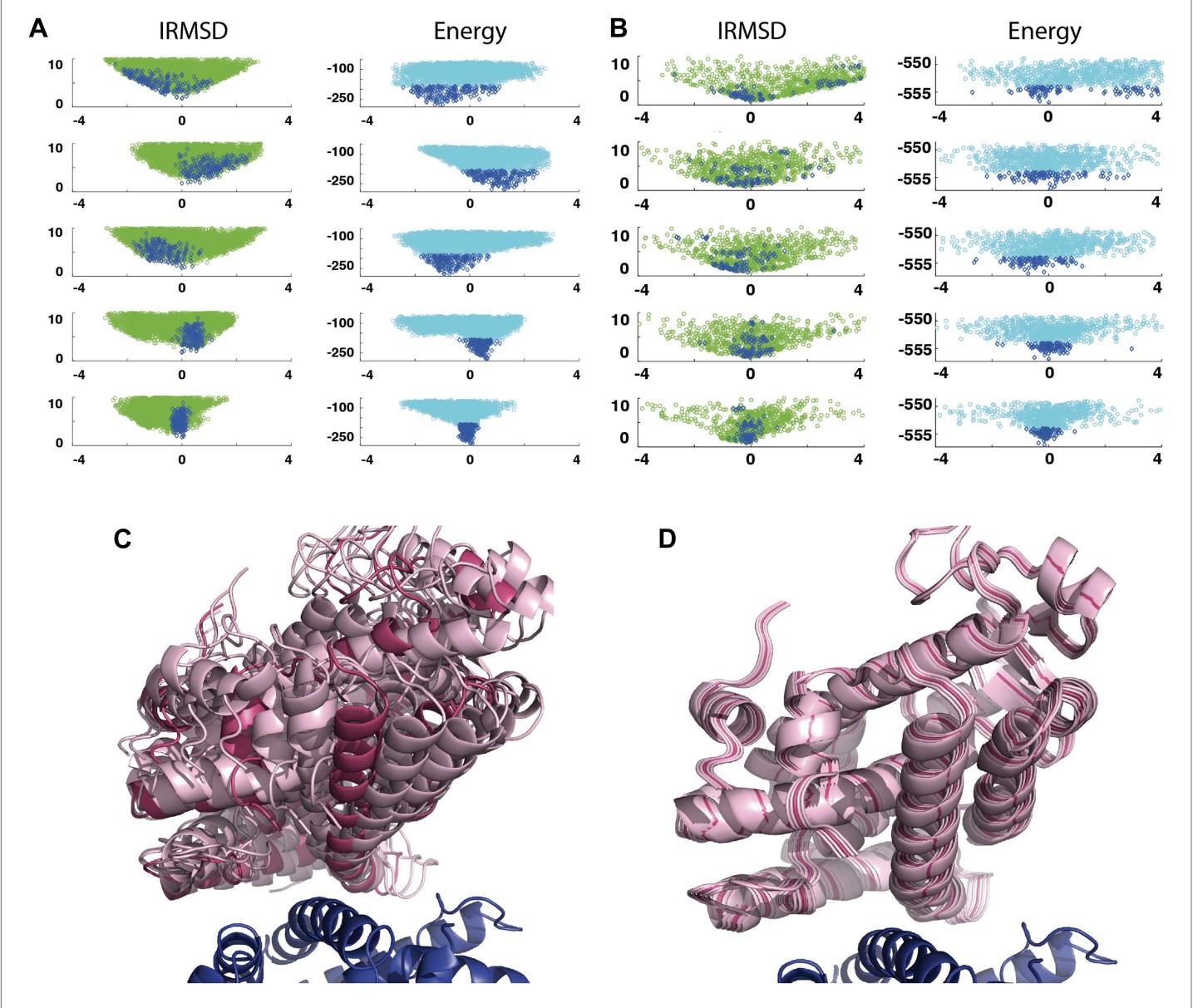

**Figure 3**. Shape of the energy landscape along the five PCA eigenvectors for the complex of PPAR-γ and RXR-α (PDB code 1K74). (**A**) Distributions of IRMSD (green) and energy (cyan) values based on structures generated by PIPER as functions of the 'balanced' coordinates shown on the x-axis. Dark blue diamonds indicate low energy data points used for the PCA. The IRMSD (y-axis in the left column) is given in Å. The energy values (on the y-axis in the right column) are given by the PIPER scoring function. (**B**) Same as *Figure 3A*, but based on structures generated by RosettaDock. The energy values (on the y-axis in the right column) are given by the RosettaDock scoring function. (**C**) Encounter complexes along the most permissive direction **v**₁. The ensemble includes mostly translations from the native state. (**D**) Encounter complexes along the most restrictive direction **v**₅.

The following figure supplements are available for figure 3:

**Figure supplement 1**. Helix H12 of PPARγ with residues of the hydrophobic patch indicated.

X-receptor α (RXRα) and the peroxisome proliferator-activated receptor γ (PPARγ), considered here as the receptor and the ligand, respectively. The PDB entry of the complex is 1K74, but we docked the unbound (separately crystallized) RXRα and PPARγ structures rather than the components from the complex. *Figure 3B* shows the distributions of the same quantities, but based on the low energy structures generated by RosettaDock. The largest eigenvalue, $\lambda_1$, is close to 50% for both energy functions.

The corresponding movements along $v_1$ are rotations of helix H12 of PPARγ around a hydrophobic patch, formed by the side chains of F432, A433, and L436, which binds to a large hydrophobic pocket of RXRα and remains almost at the same position in all low energy encounter complexes (*Figure 3C*, *Figure 3—figure supplement 1*; *Video 1*). As helix H12 rotates, the entire PPARγ moves with it until a loop formed by PPARγ residues 394 to 403 reaches a favorable position on the surface of RXRα. We note that hydrophobic patch on helix H12 and the residues connecting it to the rest of the protein (residues 413–433) are known to be essential for forming the heterodimer (*Chan and Wells, 2009*). Along the eigenvector $v_2$ that correspond to the second largest eigenvalue $λ_2$ we can observe how the amino end of helix H12 with the hydrophobic patch on it moves into its binding pocket (*Video 2*). Based on the eigenvalues $λ_1$ and $λ_2$ (*Table 1*), over 75% of all movement of PPARγ approaching RXRα occurs in the subspace spanned by the eigenvectors $v_1$ and $v_2$. Thus, this subspace can be regarded as the essential consensus of a very large number of association trajectories. In contrast to the permissive subspace, changes are very small along $v_5$ (*Figure 3D*). Since the higher energy structures (not included in the data considered for PCA) can be substantially further from the native state than the ones with low energy, we conclude that the valley based on energy is much narrower than the valley based on geometry.

As a second example, we show IRMSD and energy distributions and PCA results for an enzyme–inhibitor complex, subtilisin Carlsberg and its protein inhibitor, OMTKY3 (*Figure 4*). The PDB entry of the complex is 1R0R. For this pair of proteins, the low energy encounter complexes along the eigenvectors $v_4$ and $v_5$ show even narrower distributions than in the previous example, both for PIPER and RosettaDock (*Figure 4A,B*). Since the essentially planar inhibitor loop (residues 13 to 19 of OMTKY3) is locked into the crevice at the enzyme's active site, we expected that the motion along the most permissive direction would be the rigid body rotation of the entire inhibitor, possibly with slight readjustments of the loop. However, we have found that the motion along $v_1$ is the move of the loop, and particularly the primary specificity residue L18, deeper into the binding pocket of the enzyme (*Figure 4C*, *Figure 4—figure supplement 1*; *Video 3*). The rotation along the loop is also present, but along the eigenvector $v_2$ rather than $v_1$ (*Video 4*). Based on the eigenvectors $λ_1$ and $λ_2$ (*Table 1*), 81.8% of the movements of OMTKY3 upon binding occurs in the subspace spanned by eigenvectors $v_1$ and $v_2$ for this complex. In contrast, the motion along the most restrictive direction $v_5$ is a very small translation along the bottom of the active site (*Figure 4D*). It is important to note that, in principle, small eigenvalues identified by PCA could have also occur by chance due to undersampling a subspace. We performed simple Monte Carlo analyses to exclude this possibility ('Materials and methods').

As shown by the eigenvalues in *Table 1* and by *Figures 3 and 4*, the energy funnels derived from the PCA of the energy landscapes generated by PIPER and RosettaDock slightly differ. This is not surprising, because we specifically selected two docking programs that are very different both in terms of their sampling algorithms and scoring functions ('Materials and methods'). PIPER performs systematic rigid body sampling on a dense grid using a 'smooth' potential that allows for some

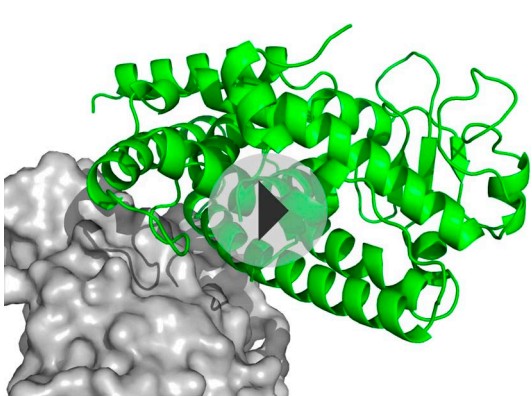

**Video 1**. Movement of PPARγ, shown as green cartoon, along the most permissive eigenvector v₁. The receptor, RXRα, is shown as grey surface.

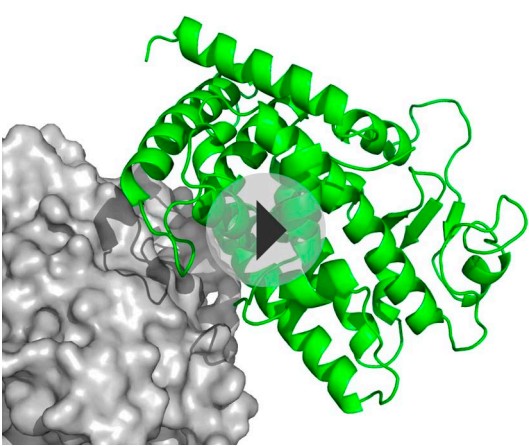

**Video 2**. Movement of PPARγ, shown as green cartoon, along the second most permissive eigenvector v₂. The receptor, RXRα, is shown as grey surface.

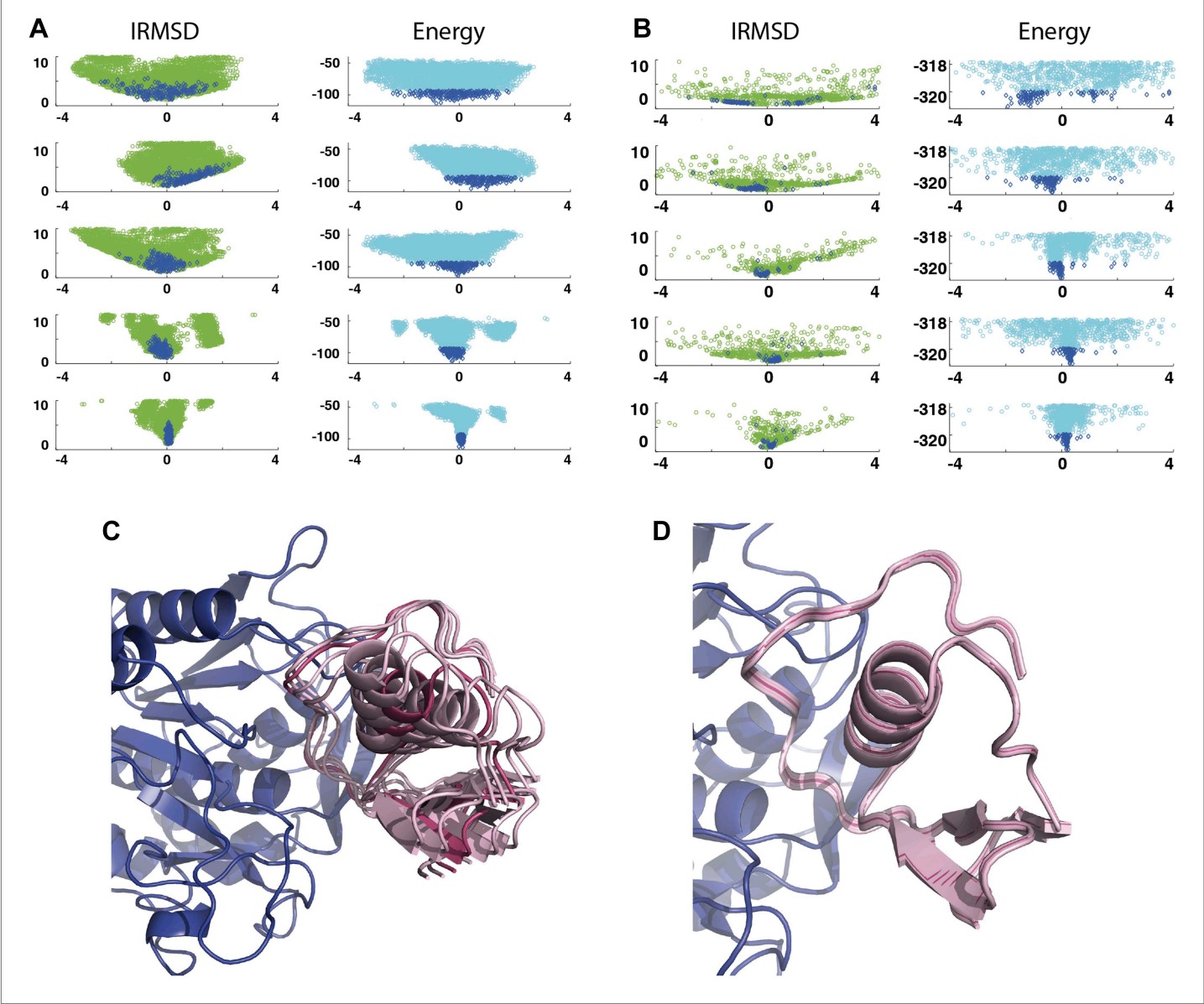

**Figure 4.** Shape of the energy landscape along the five PCA eigenvectors for the complex of subtilisin Carlsberg and its protein inhibitor, OMTKY3. All notations are as in *Figure 3*. (**A**) Distributions of interface IRMSD and energy values based on the structures generated by PIPER. (**B**) Same as *Figure 4A*, but based on the RosettaDock dataset. (**C**) Encounter complexes along the most permissive direction $v_1$. The ensemble consists of small rotations that leave the inhibitory loop position largely invariant. (**D**) Encounter complexes along the most restrictive direction $v_5$.

The following figure supplements are available for figure 4:

**Figure supplement 1.** Movement of the OMTKY3 inhibitory loop into the active site of subtilisin Carlsberg.

overlaps (*Kozakov et al., 2006*). In contrast, RosettaDock samples the region of interest using a Monte Carlo minimization algorithm, which biases the search toward low energy regions, and thus the sampling is less exhaustive than the systematic sampling by PIPER. The method periodically rebuilds the complete set of interface side chains, followed by the optimization of the rigid body displacement. The energy is locally minimized in every iteration cycle of a Monte Carlo search algorithm (*Gray et al., 2003*), and since the clashes are continuously removed, RosettaDock can use an energy function that is more sensitive to small changes in the coordinates than the energy function used in PIPER. Accordingly, Panels A and B of *Figures 3 and 4* display somewhat different shapes of the

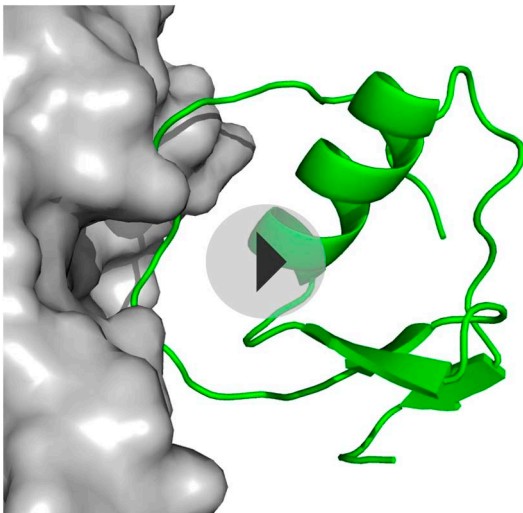

**Video 3**. Movement of the protein inhibitor, OMTKY3, shown as green cartoon, along the most permissive eigenvector $v_1$. The receptor, subtilisin Carlsberg, is shown as grey surface.

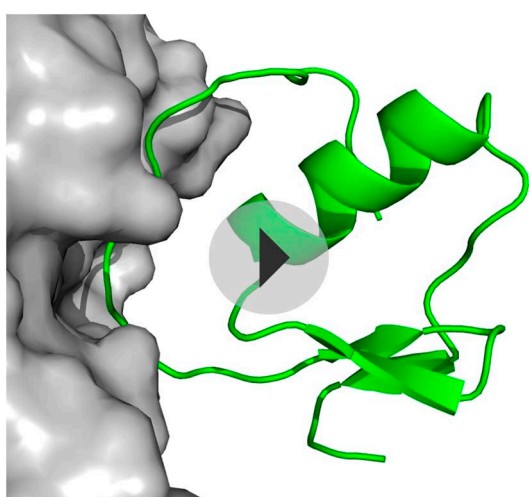

**Video 4**. Movement of the protein inhibitor, OMTKY3, shown as green cartoon, along the second most permissive eigenvector $v_2$. The receptor, subtilisin Carlsberg, is shown as grey surface.

energy distributions around the native state placed at the center of the coordinate system. Using rigid structures without local minimization, the minimum of the PIPER energy may be somewhat shifted from the native state, resulting in a more skewed energy landscape than the one obtained with RosettaDock, which generally places the energy minimum closer to the native structure and hence has a more symmetric energy landscape. In view of the differences between the two energy functions we consider it important that the PCA analyses of low energy structures generated by either PIPER or RosettaDock yield similar distributions of the eigenvalues for all 42 complexes. For each complex, both programs result in two small eigenvalues $\lambda_5$ and $\lambda_4$. Although these eigenvalues are generally somewhat smaller for PIPER, because the rigid body approximation leads to a steeper increase in energy as we move away from the minimum along the most restrictive direction, both programs clearly indicate the existence of a restrictive subspace. In addition, Monte Carlo studies, described in the 'Materials and methods', show that the restrictive subspaces predicted PIPER and RosettaDock are fairly similar. The similarity of these subspaces obtained by two very different energy functions for all 42 complexes indicates that the reduction of dimensionality is an inherent property of protein–protein association. Both programs predict that, on the other end of the spectrum, the two largest eigenvalues, $\lambda_1$ and $\lambda_2$, together exceed 75% of the total variance for most complexes. Thus, in a neighborhood of the native state the encounter complexes are essentially restricted to a two dimensional permissive subspace in the rotational/translational space, and this conclusion is independent of the docking program used.

## Discussion

### Generation of encounter complexes by docking

Assuming moderate conformational changes and using grid approximation, the FFT based global and systematic sampling of the configurational space of two interacting proteins using a physics based energy function converts the docking problem into an exactly solvable problem of statistical mechanics (*Kozakov et al., 2013*). According to the CAPRI community-wide protein–protein docking experiment, this type of approximation gives good results for a large fraction of complexes (*Lensink and Wodak, 2013*). However, it has been well known that, for most protein pairs, such global search yields low energy structures in several regions of the conformational space, some of which are far from the structure of the native complex. Physics-based energy functions are expected to be globally valid for modeling interactions between proteins, including the non-native states. Thus, one can assume the energy values that are low relative to the average energy but still exceed the energy at the global minimum may lead to the formation of relatively short-lived encounter complexes along the association pathways. As shown in this paper, the

agreement between experimental PRE data and theoretical PRE profiles calculated from the ensemble of structures generated by docking confirms this hypothesis, and thus structures of encounter complexes can be obtained simply as byproducts of docking without any further computational expense. While this result is not unexpected, in view of the limited structural information available on encounter complexes it is potentially significant.

## Identification of encounter complexes using PRE

To detect intermediate structures in the association of proteins EIN and HPr, paramagnetic labels were introduced at two sites on HPr, one at a time, and the transverse paramagnetic relaxation enhancement (PRE) rates, $\Gamma_2$, of the backbone amide protons ($^1H_N$) of EIN were measured. Since the population of the intermediate structures is generally much lower than the population of the native complex, it is important to discuss why PRE can detect the presence of encounter complexes. The major factor is that the magnitude of the PREs is proportional to $< r^{-6}>$, where r is the distance between the nucleus of interest and the paramagnetic center, and $<>$ denotes averaging over the ensemble of structures. Due to the large magnetic moment of an unpaired electron, the effect is detectable for sizeable separations (up to ~34 Å for $Mn^{2+}$). A hypothetical example can be used to explain why the method can detect states with very low populations. We consider an ensemble that includes a major species A with the population, $p_A$, of 99%, and with a paramagnetic center to proton distance of 30 Å, and a minor species B with the population, $p_B$, of 1%, and with a paramagnetic center to proton distance of 8 Å. We calculate $\Gamma_2$ for this proton in a two-site exchange system between A and B, where $\Gamma_2$ is defined as the difference in the transverse relaxation rates of the paramagnetic and diamagnetic states (*Iwahara and Clore, 2006*). For a ~30-kDa complex, for species A the $^1H$-$\Gamma_2$ arising from $Mn^{2+}$ is ~2 $s^{-1}$ ($\Gamma_{2,A}$), and for species B it is ~5.6 × $10^3$ $s^{-1}$ ($\Gamma_{2,B}$). Considering B as a short-lived encounter complex and A as the native state, and assuming that the system is in the fast exchange regime, the apparent PRE rate, $\Gamma_2$, is the population weighted average of the $\Gamma_2$ rates of the two species, that is, $\Gamma_2 = p_A \Gamma_{2,A} + p_B \Gamma_{2,B}$ (*Iwahara and Clore, 2006*). Based on this expression $\Gamma_2$ is ~30-fold larger than $\Gamma_{2,A}$, thereby permitting one to both infer the presence of, and obtain some structural information on, the minor species, because the PRE is a highly distance-dependent quantity. Thus, according to this simple explanation, the PREs can clearly capture the footprint of minor species that exchange rapidly with the native complex, in spite of their much lower concentration.

In a realistic protein–protein association the PRE rate, $\Gamma_2$, is the population weighted average of the $\Gamma_2$ rates over the native state and the entire ensemble of encounter complexes. The strong distance dependence of $\Gamma_2$ implies that the observed values are sensitive even to small conformational changes that may occur, for example, due to changes in the rotameric state of the EDTA-$Mn^{2+}$ paramagnetic probe. Thus, as shown in *Figure 2*, the PRE data, while sensitive to the presence of minor species, are also fairly noisy ('Materials and methods'). In spite of their substantial variance, the data are informative, since PREs generally also occur at residues that are far from the paramagnetic label in the native complex but are getting closer to it in some members of the encounter ensemble, clearly indicating the presence of non-native transition states. As the examples studied in this paper show, the minor species can be better detected if the label is far from the interface. In fact, a label placed close to interface generates a strong PRE signal, and thus the PRE profile back-calculated from the native structure already correlates well with the data. However, accounting for the encounter complex ensemble most likely improves the prediction even in such cases, but the improvement is generally smaller than the one with the paramagnetic label placed far from the interface.

## Reduction of dimensionality in protein–protein association

The reduction of dimensionality in molecular association was originally proposed to explain high binding rates (*von Hippel and Berg, 1989*), particularly the ability of proteins to locate their target sites along DNA (*Riggs et al., 1970*). Dimensionality reduction is caused by interaction forces that are non-specific and thus do not lead to binding at a specific site, but keep the macromolecules in proximity for a prolonged time, allowing an extensive search of the surface along certain directions while restraining the search along others (*Ubbink, 2009*). This is clearly the case for DNA, whose negative charge attracts positively charged proteins without providing a specific interaction site (*Iwahara et al., 2006*; *Gorman and Greene, 2008*). It is well known that long-range electrostatic interactions can also increase the rates of association of two proteins with net opposite charges or with strong charge dipoles, as the search for the reactive patches is facilitated by dipolar pre-orientation of the proteins

upon their approach (*Schreiber et al., 2009*). Such charge interactions prolong the lifetime of the transition state and increase the fraction of productive complexes, and thus can reduce dimensionality. However, it is frequently assumed that, due to specific charge–charge interactions and their irregular surface, proteins do not have ensembles of orientations having similar energies and thus allowing for search along the surface. We have shown here that this is definitely not the case because the energy landscape of interacting proteins, at least within the 10 Å IRMSD neighborhood of the native state, always includes a permissive subspace along which the conformation of the complex can substantially change without crossing significant energy barriers. Thus, there is no reason to assume that the interactions are nonspecific in protein-DNA association but are specific when two proteins associate. In fact, for all 42 protein pairs, some of which have strong electrostatic interactions, the energy landscape is smooth funnel in a two dimensional permissive subspace. In all cases this subspace captures at least 75% of the total motion as the two molecules approach the native state. For each of the 42 complexes we also detect a high energy subspace, which reduces the dimensionality of the space available to encounter complexes along the association pathways. Thus, there is much less difference between protein-DNA and protein–protein association than it was previously believed.

Finally we note that the reduced dimensionality of the search space can potentially simplify docking calculations, and thus the results of PCA provide several opportunities for improving the efficiency of docking methods. First, it is well known that any type of optimization is more efficient along the principal components, as large steps can be taken along the permissive directions. This is particularly the case for second-order methods such as the Newton–Raphson optimization that uses a local quadratic approximation of the energy function to find the next minimum in each iteration (*Fletcher, 1981*). It is also important that such methods require the inversion of the Hessian matrix of the energy function, and in the case of reduced dimensionality the matrix is nearly singular, leading to numerical difficulties and loss of accuracy. Once such directions are known, the problem can be avoided by regularization methods (*Fletcher, 1981*). In fact, after developing and testing a medium-range optimization method SDU, which employs quadratic semi-definite underestimation in the 5D angular space with the exponential parameters also used in this work (*Shen et al., 2008*), we understood that the approach can be made more efficient by accounting for the reduced dimensionality of the search space and adding regularization based on PCA. Another potential use is optimally selecting perturbation vectors using the relative magnitudes of the eigenvalues in biased Monte Carlo methods (*Lee et al., 1996*).

## Materials and methods

### Global sampling by the Fast Fourier transform (FFT) correlation approach

In order to fully explore the conformational space in protein–protein association we perform exhaustive evaluation of an energy function in the discretized space of mutual orientations of the two proteins using the docking program PIPER, which is based on the Fast Fourier transform (FFT) correlation approach (*Kozakov et al., 2006*). The center of mass of the first protein, defined here as the receptor, is fixed at the origin of the coordinate system, whereas the second protein (usually the smaller of the two), defined as the ligand, is rotated and translated. The translational space is represented as a grid of 1.0 Å displacements of the ligand center of mass, and the rotational space is sampled using 70,000 rotations based on a deterministic layered Sukharev grid sequence, which quasi-uniformly covers the space. The energy expression used for the FFT based sampling includes simplified van der Waals energy $E_{vdw}$ with attractive ($E_{attr}$) and repulsive ($E_{rep}$) contributions, the electrostatic interaction energy $E_{elec}$, and a statistical pairwise potential $E_{pair}$, representing other solvation effects (*Chuang et al., 2008*):

$$E = E_{vdw} + w_2 E_{elec} + w_3 E_{pair}$$

The individual energy terms are calculated by the $E_{vdw} = E_{attr} + w_1 E_{rep}$, $E_{elec} = \sum_i \sum_j [q_i q_j / \{r^2 + D^2 \exp(-r^2/4D^2)\}^{1/2}]$, and $E_{pair} = \sum_i \sum_j \varepsilon_{ij}$, where $r$ is the distance between atoms $i$ and $j$, $D$ is an atom-type independent approximation of the generalized Born radius, and $\varepsilon_{ij}$ is a pairwise interaction potential between atoms $i$ and $j$. All energy expressions are defined on the grid. The coefficients $w_1 = 4$, $w_2 = 600$, $w_3 = 5$, weight the different contributions to the scoring function, and are based on calorimetric considerations. In order to evaluate the energy function $E$ by FFT, it must be written as a sum of correlation functions. The first two terms, $E_{vdw}$ and $E_{elec}$, satisfy this

condition, whereas $E_{pair}$ is written as a sum of a few correlation functions, using an eigenvalue-eigenvector decomposition (*Kozakov et al., 2006*). For each rotation, this expression can be efficiently calculated using $P$ forward and one inverse Fast Fourier transforms. The calculations are performed for each of the 70,000 rotations, and one or several lowest energy translations for each rotation are retained. The results are clustered with a 10 Å IRMSD radius around the native coordinate.

## Generating encounter complexes

Unbound structures were used both for the receptor, EIN (chain A from PDB (*Berman et al., 2000*) entry 1ZYM) and for the ligand, HPr (chain P from PDB entry 2JEL). Encounter complexes were generated using the global systematic Fast Fourier Transform based docking program PIPER (*Kozakov et al., 2006*). The docking was performed with each of the three conformers of EDTA-Mn$^{2+}$ group, both at positions E5C and E32C of HPr (*Fawzi et al., 2010*). For each conformer, the 10,000 lowest energy complex structures were retained for the calculation of PRE rates.

## Transverse paramagnetic relaxation enhancement (PRE) rate calculation

To calculate the transverse PRE rates ($\Gamma_2$) from the ensemble of encounter complexes generated by the FFT based sampling we use the $N_{st}$ = 30,000 (10,000 for each of the three conformers of EDTA-Mn$^{2+}$) lowest energy structures. The observed PRE values are the PRE rates averaged over a population (*Iwahara et al., 2004*; *Tang et al., 2006*), and hence

$$< \Gamma_2(i) > = \left( \sum_j \Gamma_{ji} \right) \Big/ N_{st}$$

where $\Gamma_{ji}$ is the PRE rate for residue $i$ of the $j$th structure in the ensemble. Each individual $\Gamma_{ji}$ value is proportional to the inverse sixth power of distance $r_{ij}$ between the backbone amide proton (directly bonded to $^{15}$N) of the $i$th residue and the paramagnetic ion Mn$^{2+}$. To account for magnetic trap flexibility, we use the three state EDTA-Mn$^{2+}$ Solomon-Bloembergen approximation (*Iwahara et al., 2004*; *Tang et al., 2006*). With these assumptions the PRE rates are given by $\Gamma_{ji} = C \times \Sigma_k (r_{ikj})^{-6}$, where $C = 1.2 \times 10^{10}$ Å$^6/s$ and $r_{ikj}$ is the distance between residue $i$ and the $k$th state of EDTA-Mn$^{2+}$ in the $j$th structure in the low energy ensemble of docked configurations. The $\Gamma_{ji}$ values are limited to 90s$^{-1}$, that is, $\Gamma_{ji}$ = 90 if $\Gamma_{ji}$ ≥ 90. Introducing this threshold is based on the observation that both theoretical and experimental $\Gamma_{ji}$ values become very uncertain over this threshold because small distance variations strongly affect the result (*Kim et al., 2008*; *Fawzi et al., 2010*).

## Sampling by Monte Carlo minimization

The free energy landscape near the native complex was explored using both PIPER, based on the Fast Fourier transform (FFT) correlation approach (*Kozakov et al., 2006*) and RosettaDock, a docking program based on the Monte Carlo minimization (MCM) algorithm (*Gray et al., 2003*). In each MCM cycle, RosettaDock perturbs the position of the ligand by random translations and rotations, followed by adjusting the distance between the ligand and receptor to create a contact. Next, a fast MCM at low resolution optimizes the complex orientation with respect to features that do not depend on the explicit conformations of the side chains. Finally, the side chains are added, and an all-atom optimization locates the local minimum energy conformation. The complete set of interface side chains is repacked every eight cycles, followed by the optimization the rigid body displacement. After each move, side chain packing, and minimization, an energy score is calculated. The new position is kept or rejected according to the standard Metropolis acceptance criterion (*Gray et al., 2003*). RosettaDock uses a detailed energy function which includes van der Waals interactions with a linear term serving as the repulsive part, a solvation term based on a pairwise Gaussian solvent exclusion model, hydrogen bonding energies using an orientation-dependent empirical function, a rotamer probability term, residue–residue atom pair interactions for charged residues, and a simple electrostatic term across the protein–protein interface.

## Parameterizing the conformational space

The origin of the reference frame is placed at the center of the receptor, and the *z*-axis is directed toward the center of the interface of the native conformation. The translation of the ligand is described by the vector from the center of the receptor to the center of the ligand interface (as opposed to the center of the ligand), and the rotation of the ligand is also defined around the center of its interface. This choice is made to decouple, as much as possible, the effects of translation on the locations of the

interface atoms from those of the rotation. A translation vector $\mathbf{y} \in R^3$ can be represented by a triplet $(r,\theta,\varphi)$, where $r = \|\mathbf{y}\|$ is the distance, $\theta$ is the azimuth angle between the projection of $\mathbf{y}$ on the $xy$-plane and the $x$-axis (longitude, $0 \leq \theta < 2\pi$), and $\varphi$ is the zenith angle between the $z$-axis and the vector $\mathbf{y}$ (colatitude, $0 \leq \varphi < \pi$). As will be discussed, we consider $r$ separately, and focus on selecting appropriate parameterizations for $(\theta,\varphi)$ compounded with the rotational space $SO(3)$.

Parameterizing rotations is problematic because rotations are non-Euclidean in nature (i.e., travelling infinitely far in any direction will bring you back to your starting point an infinite number of times). Any attempt to parameterize the entire set of three degrees-of-freedom (DOF) rotations by an open subset of Euclidean space (as do Euler angles) will suffer from gimbal lock, the loss of rotational degrees of freedom, due to singularities in the parameter space (*Grassia, 1998*). Parameterizations that are themselves defined over non-Euclidean spaces (such as the set of unit quaternions embedded in $R^4$) may remain singularity-free, and thus avoid gimbal lock. Employing such parameterizations is complicated, however, since the numerical tools such as optimization and PCA assume Euclidean parameterizations; therefore we must either develop new tools whose domains are non-Euclidean, or complicate our systems by imposing explicit constraints, for example, to assure that the quaternions stay on the unit sphere (*Grassia, 1998*). In this paper we use *exponential maps* (*Park, 1995*; *Park and Ravani, 1997*; *Shen et al., 2008*) to project an Euclidean space onto the nonlinear rotational space. As an example, the simplest exponential map defines a local one-to-one correspondence between the unit circle, which is a nonlinear space (called the circle group), centered at 0 in the complex plane, and the tangent space at 1, which can be identified with the imaginary line in the complex plane. The exponential map for the circle group is given by $it \to e^{it}$, where $i = \sqrt{(-1)}$, $it$ specifies a point on the tangent line, and the exponential function projects it into the corresponding point of the circle (*Figure 5A*). Extending the map to three dimensions, the exponential map projecting the tangent plain of a 3D sphere onto the surface of the sphere is shown in *Figure 5B*.

For $(\theta,\varphi)$ the exponential coordinates are $(\sigma_1,\sigma_2) = (-\varphi \sin\theta, \varphi \cos\theta)$. For $SO(3)$, the exponential coordinates are $\omega = (\omega_1, \omega_2, \omega_3) \in R^3$, where $\omega_1$, $\omega_2$, and $\omega_3$ are elements of a skew-symmetric matrix (*Park, 1995*; *Shen et al., 2008*; *Mirzaei et al., 2012*). The vector $(\sigma_1, \sigma_2, \omega_1, \omega_2, \omega_3)$ defines the relative orientation of the two rigid proteins. Once this relative orientation is given, the binding distance $r$ along the translation vector $\mathbf{y}$ (which connects the centers of the receptor with the center of the ligand's interface) is determined by the assumptions that the two proteins are in contact but do not overlap (*Shen et al., 2008*). Complexes generated by the docking methods used in this work always have this property. In particular, given a complex structure generated by the FFT sampling approach PIPER (*Kozakov et al., 2006*), we can uniquely determine the corresponding $(\sigma_1, \sigma_2, \omega_1, \omega_2, \omega_3)$ coordinates

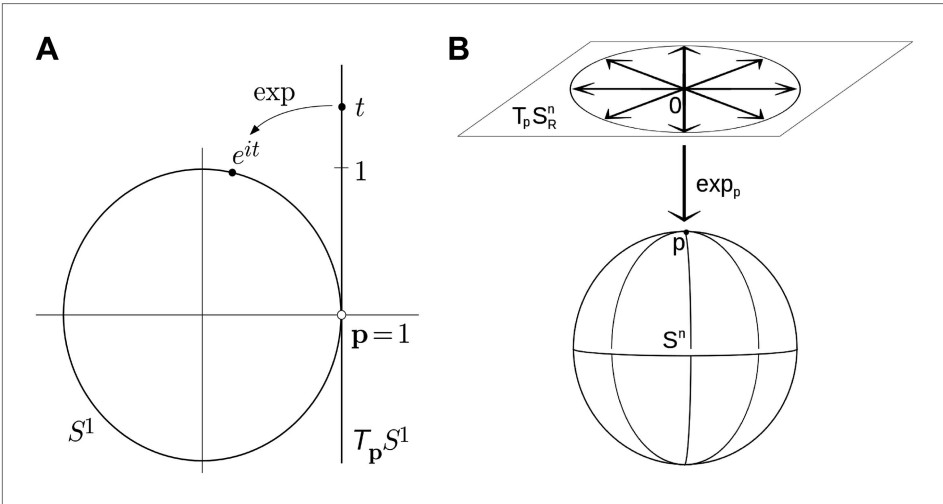

**Figure 5**. Examples of simple exponential maps. (**A**) Parameterization of the unit circle using an exponential map. The function $e^{it}$ is a local one-to-one mapping of the tangent line around p = 1 onto the unit circle. (**B**) Prameterization of the 3D unit sphere using exponential parameters.

and the value of *r*. Since RosettaDock (*Gray et al., 2003*) generally changes the conformations of the component proteins, to isolate the rigid body motion we have to fit the initial protein structures to the complex and determine the closest rigid body transformation, projecting the higher dimensional motion into the rotational-translational space.

## Sampling the near-native region and principal component analysis

We sample energy values in a sphere in the 5D parameter space around the point (0, 0, 0, 0, 0), which corresponds to the native conformation. Since we all five coordinates are angles, the radius of the sphere is also defined as an angle, and we used 22.5° as the radius, which roughly translates into a radius of 10 Å IRMSD neighborhood for relatively small proteins. After the FFT based sampling using the PIPER program (*Kozakov et al., 2006*) we selected 5000 structures, uniformly distributed within this sphere. Conformations with more than 10 Å IRMSD from the native structure were discarded, resulting in about 4000 structures. RosettaDock (*Gray et al., 2003*) was used to generate 2000 structures within the 10 Å IRMSD radius. After sampling we normalized the 5D coordinates to ensure that the variances in the sample set are the same along each coordinate axis, replacing each vector ($\sigma_1$, $\sigma_2$, $\omega_1$, $\omega_2$, $\omega_3$) with its normalized version ($x_1$, $x_2$, $x_3$, $x_4$, $x_5$). The use of such normalized coordinates removed the potential differences in the scaling of the initial angular parameters, and also rendered the results less dependent on the relative sizes of the two component proteins. For each of the two scoring functions, principal component analysis (PCA) was applied to the so-called 'most favorable' conformations whose energy values were below a certain threshold, for example, in the lowest 5% for the FFT-based sampling data. Let N denote the number of such favorable structures. The normalized coordinates ($x_1$, $x_2$, $x_3$, $x_4$, $x_5$) of the *N* structures were arranged into a *N*-by-5 matrix **X**, where each row represented a conformation. The PCA analysis computed the eigenvalues and eigenvectors of the covariance matrix

$(X–\mu_X)'(X–\mu_X)$, where $\mu_X$ is the sample mean and prime denotes transpose. We sorted the eigenvalues in descending order and normalized them such that their sum equaled 1.0 (100%). The results were denoted by $\lambda_1,…, \lambda_5$, and $v_1,…, v_5$ denote the corresponding eigenvectors, which are mutually orthogonal.

## Further validation of the energy landscape

Since small eigenvalues identified by PCA might also occur by chance due to undersampling a subspace, we performed a simple Monte Carlo analysis to show that this is not the case. PCA was based on at least 100 low energy structures for each of the 42 complexes, and thence we generated 100 random vectors in the 5D space, applied PCA to derive the eigenvalues, and performed this experiment 1000 times. Results confirmed that the probability of $\lambda_5 < 10\%$ is less than 0.01. Thus, in view of the large number of sample points we used, it is very unlikely that the small eigenvalues shown in *Table 1* occur due to undersampling.

The PCA analyses of the 5D exponential coordinates of the low energy complex conformations show clear distinctions between permissive and restrictive directions in all 42 cases. The rigid FFT-based method (*Kozakov et al., 2006*) and the Monte Carlo approach in RosettaDock (*Gray et al., 2003*) yield similar results, in spite of the fact that two methods implement fundamentally different sampling and scoring schemes. To perform a rigorous comparison of the results from the two methods we introduce the notations $\lambda_{1P}, . . . , \lambda_{5P}$ and $\lambda_{1R}, . . . , \lambda_{5R}$ for the eigenvalues based on the low energy structures generated by, respectively, PIPER and RosettaDock. Both sets of eigenvalues are ordered in descending magnitude, and $v_{1P},...,v_{5P}$ and $v_{1R},...,v_{5R}$ denote the corresponding eigenvectors. Since our main goal is to show that the restrictive subspace is largely independent of the method used, as a measure of discrepancy we will determine the angle between the restrictive subspace spanned by eigenvectors $v_{4P}$ and $v_{5P}$ based on PIPER, and the subspace spanned by $v_{4R}$ and $v_{5R}$ based on RosettaDock. As shown in the last column of *Table 1*, this angle is less than or equals to 30° for all but 5 of the complexes. Accepting that the restrictive subspaces match if they differ by less than 30° seems to be a somewhat relaxed condition. However, it is easy to show by Monte Carlo simulations that the probability of such agreement by chance for 37 of the 42 structures is negligibly small. We applied PCA to two sets of 100 random vectors in 5D, calculated the covariance matrix and its eigenvalues, for each set we selected the subspace spanned by eigenvectors corresponding to the two smallest eigenvalues, and finally determined the angle between these two 'restrictive' subspaces. By repeating this calculation 1000 times we could show that the probability of obtaining an angle below 30° is p=0.131. Although this is not a very small number, the probability that this occurs for 37 of the 42 complexes is less than $10^{-22}$. Thus, the results overwhelmingly support the claim that the restrictive

directions found by two very different methods are similar, and thus the reduction in dimensionality is an inherent property of protein–protein association. Due to the orthogonality of eigenvectors, the same similarity between PIPER and RosettaDock results also applies to the permissive subspaces spanned by the first 3 eigenvectors.

Additionally we provide Figures and Videos analogous to *Figures 3,4* and *Videos 1–4* for all 42 complexes studied in this paper (*Kozakov et al., 2014*).

# Additional information

## Competing interests

DK, DRH, DB, SV: Owns stock in Acpharis Inc which licensed FFT based sampling program PIPER for commerical use. However PIPER is free for academic and govermental research. The other authors declare that no competing interests exist.

## Funding

| Funder | Grant reference number | Author |
|---|---|---|
| National Institutes of Health | GM93147 | Dima Kozakov, Ioannis Ch Paschalidis |
| National Institutes of Health | GM61867 | Sandor Vajda |
| National Institutes of Health | Intramural program of NIDDK | G Marius Clore |
| National Science Foundation | DBI1147082 | Dima Kozakov, Sandor Vajda |
| US Israel Binational Science Foundation | 2009418 | Dima Kozakov, Ora Schueler-Furman |
| Russian Ministry of Education and Science | 14.A18.21.1973 | Dima Kozakov |

The funders had no role in study design, data collection and interpretation, or the decision to submit the work for publication.

## Author contributions

DK, KL, DRH, Conception and design, Acquisition of data, Analysis and interpretation of data, Drafting or revising the article; DB, PV, OS-F, ICP, GMC, Conception and design, Drafting or revising the article; JZ, Acquisition of data, Analysis and interpretation of data; SV, Conception and design, Analysis and interpretation of data, Drafting or revising the article

# Additional files

## Major datasets

The following dataset was generated:

| Author(s) | Year | Dataset title | Dataset ID and/or URL | Database, license, and accessibility information |
|---|---|---|---|---|
| Kozakov D, Li K, Hall D, Beglov D, Zheng J, Vakili P, Schueler-Furman O, Paschalidis I, Clore G, Vajda S | 2014 | Data from: Encounter complexes and dimensionality reduction in protein–protein association | 10.5061/dryad.98n8c | Publicly available at Dryad (http://datadryad.org/). |

The following previously published dataset was used:

| Author(s) | Year | Dataset title | Dataset ID and/or URL | Database, license, and accessibility information |
|---|---|---|---|---|
| Hwang H, Vreven T, Chen R, Mintseris J, Janin J, Weng Z | 2012 | Protein Docking Benchmark | http://zlab.umassmed.edu/zdock/benchmark.shtml | Openly available at http://zlab.umassmed.edu/zdock/benchmark.shtml. |

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
