## [Decision Letter]

Thank you for sending your work entitled “Encounter Complexes and Dimensionality Reduction in Protein-Protein Association” for consideration at *eLife*. Your article has been evaluated by a Senior editor (John Kuriyan), a member of our Board of Reviewing Editors, and 2 reviewers. The comments were then further discussed and a consensus was determined. Prior reaching a final decision about the manuscript, we need to consider your response to address the following substantial issue that emerged from those discussions:

Docking calculations were used to thoroughly sample association modes between two proteins and identify ensembles of low energy poses. These ensembles were then used to back calculate paramagnetic relaxation enhancement (PRE) profiles, which were then compared to those derived from experimental PRE measures by NMR. The results presented show that sampling by docking is consistent with the encounters measured experimentally by PRE. The novelty of this study lies with comparing the docking ensembles with ensembles derived from PRE experiments, and using the agreement with experimentally measured PRE as a means to establish the legitimacy of the docking ensembles as representing encounter complexes. Yet little attention is devoted to the PRE data themselves and how they can be related to docking poses and their energetics. How noisy are these data? How much slack is there in associating a given binding mode with a PRE signal. While this study may be the first time the funnel-like shape for the docking reaction coordinate has (some) experimental support, one should be cautious as the funnel shape is deduced from the docking calculations only. Until we understand the limitations of the PRE data it is hard to tell if they support the funnel shape. If indeed the main conclusion of the paper is that the funnel shape is supported by PRE experiments, then more must be provided about the experimental method, the uncertainty in the experimental data, and the sensitivity of the comparison between the docking results and the experiments.

Specific points to be addressed are:

1) A serious omission is the failure to include an important control where PRE back calculations are performed from docking poses with randomized energies. Would computed PRE profiles from these randomized ensembles look very different from those derived from the docking ensembles and/or experiment?

2) The Mn^2+^ probe appears to be located in only a few sites on each protein. It would be useful to show schematically where these sites actually are in Figures 2, 3 and 4. The authors should confirm that the Mn^2+^ probes really were located at E5 and E32 (it is implied but not stated), and exactly which nitrogens were labelled with ^15^N (was it all amide nitrogens)?] In this regard it would have been extremely informative to provide a more detailed description of how PRE profiles are back calculated from association modes and their energy levels, and to discuss the inherent error rate in the experimental PRE signal itself: How does the fact that PRE is 'a technique extremely sensitive to the presence of lowly populated states...' translate in practical terms? Some of these issues are succinctly mentioned in the Introduction but not fully addressed.

---

## [Author Response]

*The novelty of this study lies with comparing the docking ensembles with ensembles derived from PRE experiments, and using the agreement with experimentally measured PRE as a means to establish the legitimacy of the docking ensembles as representing encounter complexes. Yet little attention is devoted to the PRE data themselves and how they can be related to docking poses and their energetics. How noisy are these data? How much slack is there in associating a given binding mode with a PRE signal. While this study may be the first time the funnel-like shape for the docking reaction coordinate has (some) experimental support, one should be cautious as the funnel shape is deduced from the docking calculations only. Until we understand the limitations of the PRE data it is hard to tell if they support the funnel shape. If indeed the main conclusion of the paper is that the funnel shape is supported by PRE experiments, then more must be provided about the experimental method, the uncertainty in the experimental data, and the sensitivity of the comparison between the docking results and the experiments*.

We have made three main changes to address the above general criticism as follows: 1) In the Results section we added a new subsection entitled “PRE experiments and theoretical PRE profiles based on structure”. In this subsection we provide more information on the paramagnetic relaxation enhancement (PRE) experiment and its implications (see our response to the specific comments below).

2) Figure 2 now includes the error bars of the experimental PRE data points.

3) We added a new section to the Discussion, “Identification of encounter complexes using PRE”. In this section we first provide an example explaining why PRE data are capable of indicating the presence of transition states, in spite of their low population. Second, we show that although the PRE data are inherently noisy due to the strong distance dependence, they still can unambiguously reveal if minor species are present.

*Specific points to be addressed*
*are*:

*1) A serious omission is the failure to include an important control where PRE back calculations are performed from docking poses with randomized energies. Would computed PRE profiles from these randomized ensembles look very different from those derived from the*
*docking ensembles and/or experiment*?

We thank the reviewers for suggesting this control experiment. We have followed the advice, and to demonstrate the importance of the energy function when generating the encounter complexes performed docking calculations using a scoring function without long-range energy terms, i.e., considering only the attractive and repulsive components of the van der Waals energy. This simplified energy function yields docked structures that have good shape complementarity, but have no favorable electrostatic or chemical interactions (fourth paragraph of the Results section entitled “PRE experiments and theoretical PRE profiles based on structure”). The 30,000 structures with the lowest van der Waals energy from this “shape-complementarity only” docking were then used for back-calculating theoretical PRE profiles. The results of these calculations, shown in the new Figure 2—figure supplement 1, make absolutely clear the PRE rates based on the ensemble of structures generated without a proper energy function do not show any resemblance to the observed PRE data. We also show that the correlation coefficients between the experimental PRE rates and the theoretical PRE profiles based on these docked structures are negative. We think that these control calculations convincingly show the need for using a meaningful physics based energy function in order to predict the observed PRE data.

*2) The Mn*^*2+*^
*probe appears to be located in only a few sites on each protein. It would be useful to show schematically where these sites actually are in*
Figures 2, 3 and 4*. The authors should confirm that the Mn*^*2+*^
*probes really were located at E5 and E32 (it is implied but not stated), and exactly which nitrogens were labelled with*
^15^*N (was it all amide nitrogens)?] In this regard it would have been extremely informative to provide a more detailed description of how PRE profiles are back calculated from association modes and their energy levels, and to discuss the inherent error rate in the experimental PRE signal itself: How does the fact that PRE is 'a technique extremely sensitive to the presence of lowly populated states...' translate in practical terms? Some of these issues are succinctly mentioned in the Introduction but not fully addressed*.

In two new subsections added to the revised manuscript we address the above points as follows:

A) The subsection “PRE experiments and theoretical PRE profiles based on structure” in the Results provides a more detailed description of the PRE titration experiment than in the original submission, and hopefully clarifies the issues raised by the reviewers.

B) In the subsection “Identification of encounter complexes using PRE” (Discussion) we (1) discuss the sources of errors in the PRE data; (2) show that due to its strong distance dependence, PRE is extremely sensitive to the presence of lowly populated states; and (3) explain that although the method is inherently noisy due to the strong distance dependence, it unambiguously reveals if minor species are present.